# DISENTANGLING NEURAL MECHANISMS
# FOR PERCEPTUAL GROUPING

**Junkyung Kim**[†][‡]**, Drew Linsley**[†]**, Kalpit Thakkar & Thomas Serre**
Department of Cognitive, Linguistic and Psychological Sciences
Brown University
Providence, RI 02912, USA
`junkyung@google.com`
`{drew_linsley,kalpit_thakkar,thomas_serre}@brown.edu`

## ABSTRACT

Forming perceptual groups and individuating objects in visual scenes is an essential step towards visual intelligence. This ability is thought to arise in the brain from computations implemented by bottom-up, horizontal, and top-down connections between neurons. However, the relative contributions of these connections to perceptual grouping are poorly understood. We address this question by systematically evaluating neural network architectures featuring combinations bottom-up, horizontal, and top-down connections on two synthetic visual tasks, which stress low-level "Gestalt" vs. high-level object cues for perceptual grouping. We show that increasing the difficulty of either task strains learning for networks that rely solely on bottom-up connections. Horizontal connections resolve straining on tasks with Gestalt cues by supporting incremental grouping, whereas top-down connections rescue learning on tasks with high-level object cues by modifying coarse predictions about the position of the target object. Our findings dissociate the computational roles of bottom-up, horizontal and top-down connectivity, and demonstrate how a model featuring all of these interactions can more flexibly learn to form perceptual groups.

## 1 INTRODUCTION

The ability to form perceptual groups and segment scenes into a discrete set of objects constitutes a fundamental component of visual intelligence. Decades of research in biological vision have suggested that biological visual systems extract these objects through visual routines for perceptual grouping, which rely on feedforward (or bottom-up) and potentially feedback (or recurrent) processes (Roelfsema, 2006; Roelfsema & Houtkamp, 2011; Wyatte et al., 2014). Feedforward processes group scenes by encoding increasingly more complex feature conjunctions through a cascade of filtering, rectification and normalization operations. As shown in Fig. 1a, these computations can be sufficient to detect and localize objects in scenes with little or no background clutter, or when an object "pops out" because of color, contrast, etc. (Nothdurft, 1991). However, as illustrated in Fig. 1b, visual scenes are usually complex and contain objects interposed in background clutter. When this is the case, feedforward processes alone are often insufficient for perceptual grouping (Herzog & Clarke, 2014; Pelli et al., 2004; Freeman et al., 2012; Freeman & Pelli, 2010), and it has been suggested that our visual system leverages feedback mechanisms to refine an initially coarse scene segmentation (Ullman, 1984; Roelfsema & Houtkamp, 2011; Treisman & Gelade, 1980; Lamme & Roelfsema, 2000).

To rough approximation, there are two distinct types of perceptual grouping routines that are associated with feedback computations. One routine involves grouping low-level visual features with their neighbors according to Gestalt laws like similarity, good continuation, etc. (Fig. 1b, top; Jolicoeur et al., 1986; 1991; Pringle & Egeth, 1988; Roelfsema et al., 1999; Houtkamp & Roelfsema, 2010; Houtkamp et al., 2003; Roelfsema et al., 2002). Another routine is object-based and mediated by

---

[†]These authors contributed equally to this work.
[‡]DeepMind, London, UK.

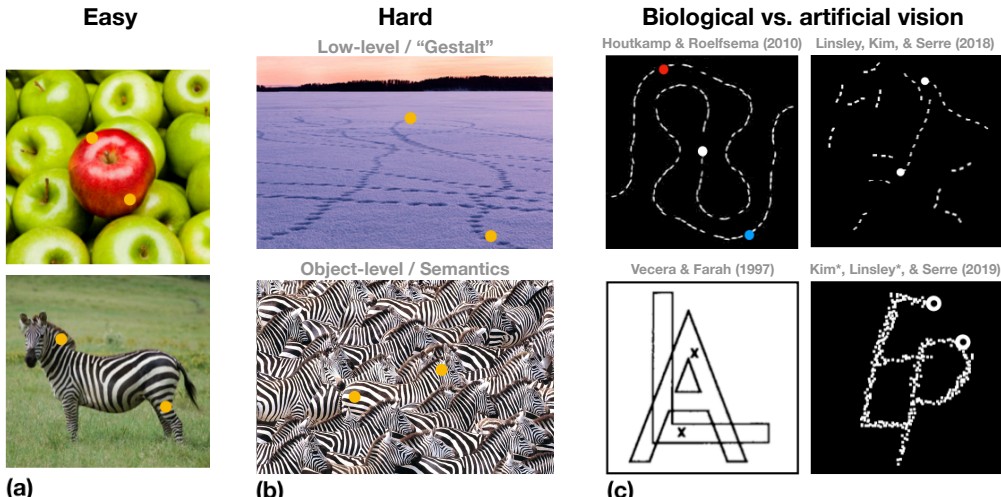

Figure 1: **"Are both dots on the same object?"** **(a) Bottom-up grouping.** Our visual system can segregate an object from its background with rapid, bottom-up mechanisms, when the object is dissimilar to its background. In this case, the object is said to "pop-out" from its background, making it trivial to judge whether the two dots are on the same object surface or not. **(b) Non-bottom-up processes.** Perceptual grouping mechanisms help segment a visual scene. **(b, Top)** The elements of a behaviorally relevant perceptual object, such as a path formed by tracks in the snow, are transitively grouped together according to low-level "Gestalt" principles. This allows the observer to trace a path from one end to the other. **(b, Bottom)** Alternatively, observers may rely on prior knowledge or semantic cues to segment an object from a cluttered background. **(c) Synthetic grouping tasks. (c, Top)** The Pathfinder challenge (reproduced with permission from Linsley et al. (2018b) and inspired by Houtkamp & Roelfsema 2010) involves answering a simple question: "Are the two dots connected by a path?" This task can be easily solved with Gestalt grouping strategies. **(c, Bottom)** Here, we introduce a novel "cluttered ABC" (cABC) challenge, inspired by Vecera & Farah (1997). The cABC challenge asks the same question on visual stimuli for which Gestalt strategies are ineffective. Instead, cABC taps into object-based strategies for perceptual grouping.

high-level expectations about the shape and structure of perceptual objects. In this routine, feedback refines a coarse initial feedforward analysis of a scene with high-level hypotheses about the objects it contains (Fig. 1b, bottom; Vecera & Farah, 1997; Vecera & O'Reilly, 1998; Vecera, 1993; Zemel et al., 2002). Both of these feedback routines are iterative and rely on recurrent computations.

What are the neural circuits that implement Gestalt vs. object-based routines for perceptual grouping? Visual neuroscience studies have suggested that these routines emerge from specific types of neural interactions: (i) horizontal connections between neurons within an area, spanning spatial locations and potentially feature selectivity (Stettler et al., 2002; Gilbert & Wiesel, 1989; McManus et al., 2011; Bosking et al., 1997; Schmidt et al., 1997; Wannig et al., 2011), and (ii) descending top-down connections from neurons in higher-to-lower areas (Ko & von der Heydt, 2018; Gilbert & Li, 2013; Tang et al., 2018; Lamme et al., 1998; Murray et al., 2004; 2002). The anatomical and functional properties of these feedback connections have been well-documented (see Gilbert & Li 2013 for a review), but the *relative* contributions of horizontal vs. top-down connections for perceptual grouping remains an open question.

**Contributions** Here we investigate the function of horizontal vs. top-down connections during perceptual grouping. We evaluate each network on two perceptual grouping "challenges" that are designed to be solved by either Gestalt or object-based visual routines. Our contributions are:

- We introduce a biologically-inspired deep recurrent neural network (RNN) architecture featuring feedforward and feedback connections (the TD+H-CNN). We develop a range of lesioned variants of this network for measuring the relative contributions of feedforward

and feedback processing by selectively disabling either horizontal (TD-CNN), top-down (H-CNN), or both types of connections (BU-CNN).

- We introduce the cluttered ABC challenge (cABC), a synthetic visual reasoning challenge that requires object-based grouping strategies (Fig. 1c, bottom-right). We pair this dataset with the Pathfinder challenge (Linsley et al. 2018b, Fig. 1c, top-right) which features low-level Gestalt cues. Both challenges allow for a parametric control over image variability. With these challenges, we investigate whether each of our models can learn to leverage either object-level of Gestalt cues for grouping.

- We find that top-down connections are key for grouping objects when semantic cues are present, whereas horizontal connections support grouping when they are not. Of the network models tested (which included ResNets and U-Nets), only our recurrent model with the full array of connections (the TD+H-CNN) solves both tasks across levels of difficulty.

- We compare model predictions to human psychophysics data on the same visual tasks and show that human judgements are significantly more consistent with image predictions from our TD+H-CNN than ResNet and U-Net models. This indicates that the strategies used by the human visual system on difficult segmentation tasks are best matched by a highly-recurrent model with bottom-up, horizontal, and top-down connections.

## 2 RELATED WORK

**Recurrent neural networks** Recurrent neural networks (RNNs) are a class of models that can be trained with gradient descent to approximate discrete-time dynamical systems. RNNs are classically featured in sequence learning, but have also begun to show promise in computer vision tasks. One example of this includes autoregressive RNNs that learn the statistical relationship between neighboring pixels for image generation (Van Den Oord et al., 2016). Another approach, successfully applied to object recognition and super-resolution tasks, is to incorporate convolutional kernels into RNNs (O'Reilly et al., 2013; Liang & Hu, 2015; Liao & Poggio, 2016; Kim et al., 2016; Zamir et al., 2017). These convolutional RNNs share kernels across processing timesteps, allowing them to achieve processing depth with a fraction of the parameters needed for a CNN of equivalent depth. Such convolutional RNN models have also been used by multiple groups as a foundation for vision models with biologically-inspired feedback mechanisms (Lotter et al., 2016; Spoerer et al., 2017; Kietzmann et al., 2019; Wen et al., 2018; Linsley et al., 2018b; Nayebi et al., 2018a; Kar et al., 2019).

Here, we construct convolutional-RNN architectures using the feedback gated recurrent unit (fGRU) proposed by Linsley et al. (2018a). The fGRU extends the horizontal gated recurrent unit (hGRU, Linsley et al. 2018b), to implement (i) horizontal connections between units in a processing layer separated by spatial location and/or feature channel, and (ii) top-down connections from units in higher-to-lower network layers. The fGRU was applied to contour detection in natural and electron microscopy images, where it was found to be more sample efficient than state-of-the-art models (Linsley et al., 2020). We leverage a modified version of this architecture to study the relative contributions of horizontal and top-down interactions for perceptual grouping.

**Synthetic visual tasks** There is a long history of using synthetic visual recognition challenges for evaluating computer vision algorithms (Ullman, 1996; Fleuret et al., 2011). There is a popular dataset of simple geometric shapes used to evaluate instance segmentation algorithms (featured in Reichert & Serre 2014; Greff et al. 2016), and variants of the cluttered MNIST dataset and CAPTCHA datasets have been helpful for testing model tolerance to visual clutter (e.g., Gregor et al. 2015; Sabbah et al. 2017; Spoerer et al. 2017; George et al. 2017a; Jaderberg et al. 2015; Sabour et al. 2017). Recent synthetic challenges have sought to significantly ramp up task difficulty (Michaelis et al., 2018), and/or to parameterically control intra-class image variability (Kim et al., 2018; Linsley et al., 2018b). The current work follows this trend of developing novel synthetic stimuli that are tightly controlled for low-level biases (unlike e.g., MNIST), and to design parametric controls (e.g., intra-class variability) for task difficulty to try to strain network models.

In the current study, we begin with the "Pathfinder" challenge (Linsley et al., 2018b), which consists of images with two white markers that may – or may not – be connected by a long path (Fig. 2). The challenge illustrated how a shallow network with horizontal connections could recognize a connected path by learning an efficient incremental grouping routine, while feedforward architectures (CNNs)

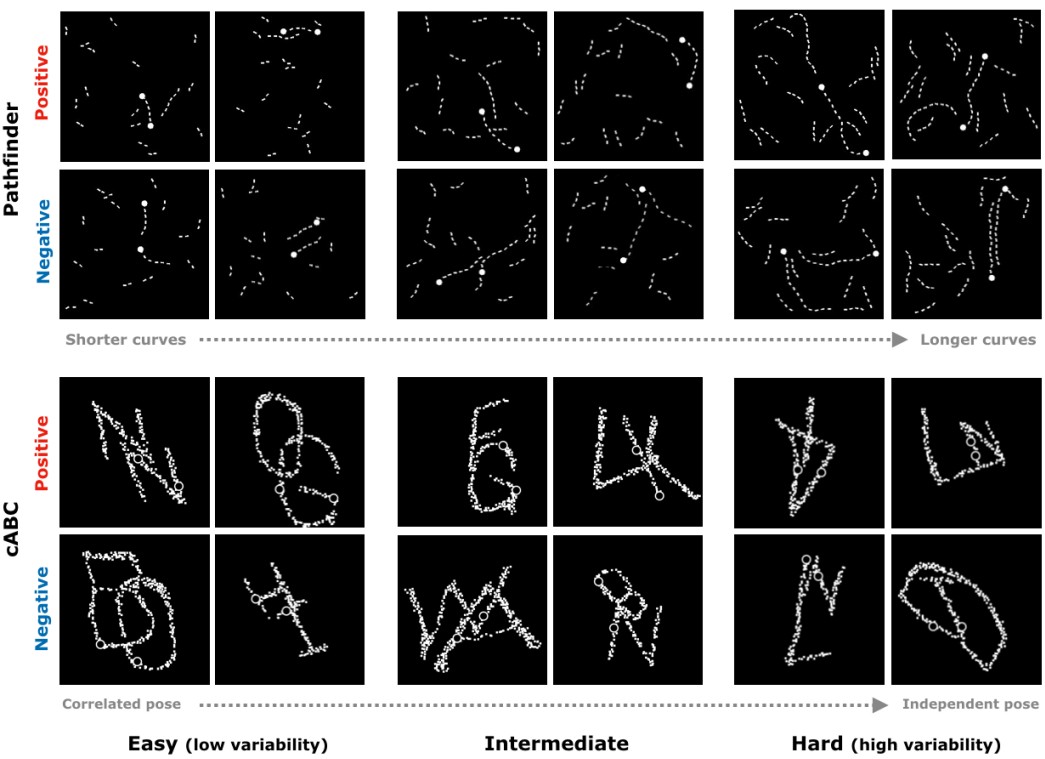

Figure 2: **An overview of the two synthetic visual reasoning challenges used: the "Pathfinder" (top two rows) and the "cluttered ABC" (cABC, bottom two rows).** On both tasks, a model must judge whether the two white markers fall on the same or different paths/letters. For both, we parametrically controlled task difficulty by adjusting intra-class image variability in the image dataset. In the Pathfinder challenge, we increased the length of the target curve; in cABC, we decreased the correlation between geometric transformations applied to the two letters while also increasing the relative positional variability between letters.

with orders of magnitude more free parameters struggled to learn the same task. Our novel "cluttered ABC" (cABC) challenge complements the Pathfinder challenge by posing the same task on images that feature semantic objects rather than curves. Thus, cABC may be difficult to solve using the same incremental grouping routine as Pathfinder, which could capture both objects in a single group (Fig. 2).

## 3 SYNTHETIC PERCEPTUAL GROUPING CHALLENGES

Pathfinder and cABC challenges share a key similarity in their stimulus design: white shapes are placed on a black background along with two circular and white "markers" (Fig. 2). These markers are either placed on two different objects or the same object, and the task posed in both datasets is to discriminate between these two alternatives. The two challenges depict different types of objects. Images in the Pathfinder challenge contain two flexible curves, whereas the cABC challenge uses overlapping English-alphabet characters that are transformed in appearance and position.

**Local vs. object-level cues for perceptual grouping**   Differences in the types of shapes used in the two challenges make them ideally suited for different feedback grouping routines. The Pathfinder challenge features smooth and flexible curves, making it well suited for incremental Gestalt-based grouping (Linsley et al., 2018b). Conversely, the English-alphabet characters in the cABC challenge make it a good match for a grouping routine that relies on semantic-level object cues.

The two synthetic challenges are designed to cause a high computational burden in models that rely on sub-optimal grouping routines. For instance, cluttered, flexible curves in the Pathfinder challenge are well-characterized by local continuity but have no set "global shape" or any semantic-level cues. This makes the task difficult for feedforward models like ResNets because the number of fixed templates that they must learn to solve it increases with the number of possible shapes the flexible paths may take. In contrast, letters in cABC images are globally stable but locally degenerate (due to warping and pixelation; see Appendix A for details of its design.), and letters may overlap with each other. Because the strokes of overlapping letters form arbitrary conjunctions, a feedforward network needs to learn a large number of templates to discriminate real from spurious stroke conjunctions.

Overall, we consider these challenges two extremes on a continuum of perceptual grouping cues found in nature. On one hand, Pathfinder curves provide reliable Gestalt cues without any semantic information. On the other hand, cABC letters are semantic but do not have local grouping cues. As we describe in the following sections, human observers easily solve both challenges with minimal training, indicating that biological vision is capable of relying on a single routine for perceptual grouping.

**Parameterization of image variability**   It is difficult to disentangle visual strategies of neural network architecture choices using standard computer vision datasets. For instance, the relative advantages of visual routines implemented by horizontal vs. top-down connections can be washed out by neural networks latching onto trivial dataset biases. More importantly, the typical training regime for deep learning involves big datasets, and in this regime, the ability of neural networks to act as universal function approximators can make inductive biases less important.

We overcome these limitations with our synthetic challenges. Each challenge consists of three limited-size datasets of different task difficulties (easy, intermediate and hard; characterized by increasing levels of intra-class variability). We train and test network architectures on each of the three datasets and measure how accuracy changes when the need for stronger generalization capabilities arises as difficulty increases. This method, called "straining" (Kim et al., 2018; Linsley et al., 2018b), dissociates an architecture's expressiveness from other incidental factors tied to a particular image domain. Architectures with appropriate inductive biases are more likely to generalize as difficulty/intra-class variability increases compared to architectures that rely on low-bias routines that can resemble "rote memorization" (Zhang et al., 2016).

Difficulty of Pathfinder challenge is parameterized by the length of target curves in the images (6-, 9-, or 14-length; Linsley et al. 2018b). cABC challenge difficulty is controlled in two ways: first, by increasing the number of possible relative positional arrangements between the two letters. Second, by decreasing the correlation between random transformation parameters (rotation, scale and shear) applied to each letter in an image. For instance, on the "Easy" difficulty dataset, letter transformations are correlated: the centers of two letters are displaced by a distance uniformly sampled in an interval between 25 and 30 pixels, and the same affine transformation is applied to both letters. In contrast, letter transformations are independent: letters are randomly separated by 15 to 40 pixels, and random affine transformations are independently sampled and applied to each. We balance the variation of these random transformation parameters across the three difficulty levels so that the total variability of individual letter shape remains constant (i.e., the number of transformations applied to each letter in a dataset is the same across difficulty levels). Harder cABC datasets lead to increasingly varied conjunctions between the strokes of overlapping letters, making them well suited for models that can iteratively process semantic groups instead of learning fixed feature templates. See Appendix A for more details about cABC generation.

## 4   NETWORK ARCHITECTURES

We test the role of horizontal vs. top-down connections for perceptual grouping using different configurations of the same recurrent CNN architecture (Fig. 3b). This model is equipped with recurrent modules called fGRUs (Linsley et al., 2018a), which can implement top-down and/or horizontal feedback (Fig. 3b). By selectively disabling top-down connections, horizontal connections, or both types of feedback, we directly compared the relative contributions of these connections for solving Pathfinder and cABC tasks over a basic deep convolutional network "backbone" capable of only feedforward computations. As an additional reference, we test representative feedforward

architectures for computer vision: residual networks (He et al., 2016) and a U-Net (Ronneberger et al., 2015).

**The fGRU module** The fGRU takes input from two unit populations: an external drive, **X**, and an internal state, **H**. The fGRU integrates these two inputs to get an updated state for a given processing timestep **H**[$t$], which is passed to the next layer. Each layer's recurrent state is sequentially updated during a single timestep, letting the model perform bottom-up to top-down sweeps of processing. When applied to static images, timesteps of processing allow feedback connections to spread and evolve, allowing units in different spatial positions and layers to interact with each other. This iterative spread of activity is critical for implementing dynamic grouping strategies. On the final timestep of processing, the fGRU hidden state **H** closest to image resolution is sent to a readout module to compute a binary category prediction.

In our recurrent architecture (the TD+H-CNN), an fGRU module implements either horizontal or top-down feedback. When implementing horizontal feedback, an fGRU takes as external drive (**X**) the activities from a preceding convolutional layer and iteratively updates its evolving hidden state **H** using spatial (horizontal) kernels (Fig. 3b, green fGRU).

In an alternative configuration, an fGRU can implement top-down feedback by taking as input hidden state activities from two other fGRUs, one from a lower layer and the other from a high layer (Fig. 3b, red fGRU). This top-down fGRU uses a local ($1 \times 1$) kernel to first suppress the low-level $\mathbf{H}^{(\ell)}[t]$ with the activity in $\mathbf{H}^{(\ell+1)}[t]$, then a separate kernel to "facilitate" the residual activity, supporting operations like sharpening or "filling-in". Additional details regarding the fGRU module are in Appendix C.

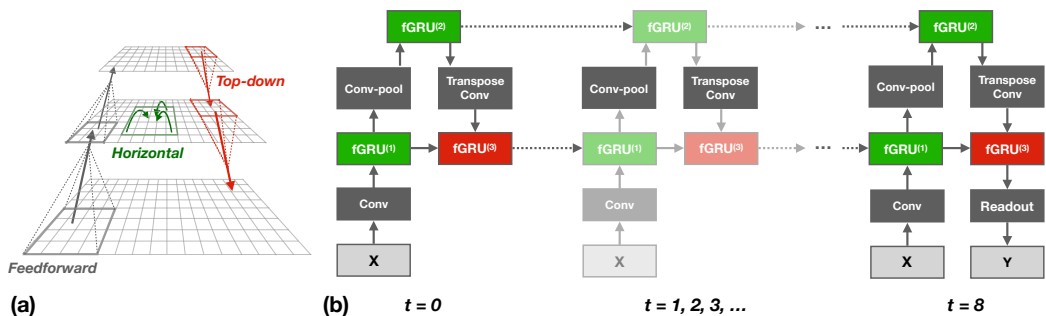

Figure 3: **We define a convolutional RNN model that can learn feedforward (gray), horizontal (green), and/or top-down (red) connections to disentangle the contributions of each type of connections for perceptual grouping. (a)** A conceptual diagram of the TD+H-CNN architecture, depicting three layers of activity and feedforward, horizontal, and top-down connections. **(b)** A deep learning diagram of the TD+H-CNN, unraveled over processing timesteps. Bottom-up to top-down passes of the input image allow recurrent horizontal (green fGRUs) and top-down (red fGRU) interactions between units to evolve. This allows the model to implement complex grouping strategies, like transitively linking target features together or selecting whole objects from clutter.

**Recurrent networks with feedback** Our full model architecture, which we call TD+H-CNN (Fig. 3b), introduces three fGRU modules into a CNN to implement top-down (TD) and horizontal (H) feedback. The architecture consists of a downsampling (Conv-pool) pathway and an upsampling (Transpose Conv) pathway as in the U-Net (Ronneberger et al., 2015), and processes images in a bottom-up to top-down loop through its entire architecture at every timestep. The downsampling pathway lets the model implement feedforward and horizontal connections, and the upsampling pathway lets it implement top-down interactions (conceptualized in Fig. 3b, red fGRU). The first two fGRUs in the model learn horizontal connections at the lowest- and highest-level feature processing layers, whereas the third fGRU learns top-down interactions between these two fGRUs.

We define three variants of the TD+H-CNN by lesioning fGRU modules for horizontal and/or top-down connections: a top-down-only architecture (TD-CNN); a horizontal-only architecture (H-CNN); and a feedforward-only architecture ("bottom-up" BU-CNN; see Appendix C for details).

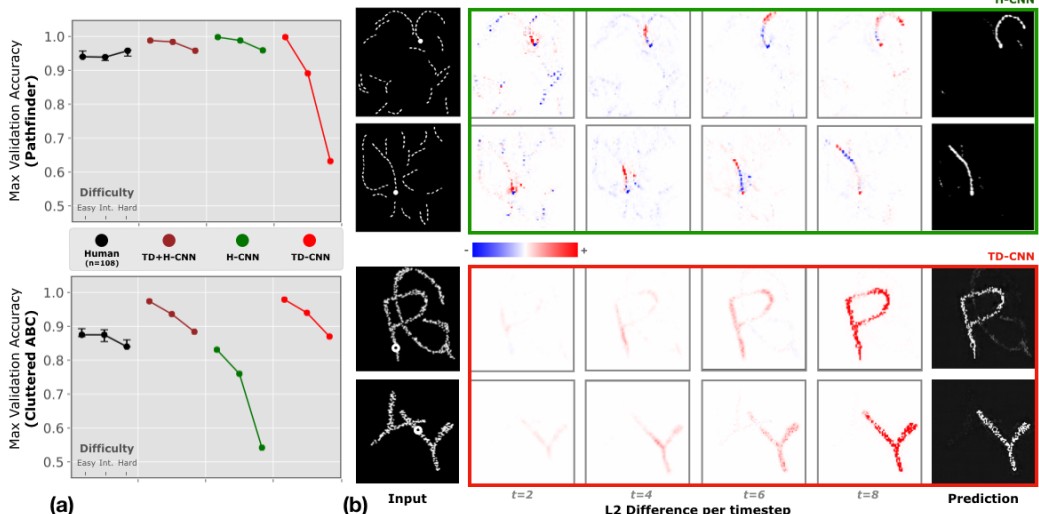

Figure 4: **Pathfinder and cABC challenges disentangle the relative contributions of horizontal vs. top-down connections for perceptual grouping. (a)** Accuracies of human and neural network models on three levels of the Pathfinder and cABC challenges. While human participants and the TD+H-CNN model are able to solve all difficulties of each challenge, Pathfinder strains the TD-CNN and cABC strains the H-CNN. **(b)** We visualized the computations learned by our networks using images containing single markers (See Appendix B for details.) We found that H-CNN and TD-CNN learn distinct strategies to solve Pathfinder and cABC challenges, respectively. The H-CNN solves Pathfinder by iteratively grouping segments of a target path while suppressing background clutter. The TD-CNN solves cABC by globally facilitating activity of a target letter while suppressing the other letter. Model strategies are visualized by plotting the per-timestep L2 difference between hidden states of the fGRU responsible for model predictions.

**Reference networks**     We also measure the performance of popular neural network architectures on Pathfinder and cABC. We tested three "Residual Network" (ResNets, He et al. 2016) with 18, 50, and 152 layers and a "U-Net" (Ronneberger et al., 2015) which uses a VGG16 encoder. Both model types utilize skip connections to mix information between processing layers. However, the U-Net uses skip connections to pass activities between layers of a downsampling encoder and an upsampling decoder. This pattern of residual connections effectively makes the U-Net equivalent to a network with top-down feedback that is simulated for one timestep.

## 5 EXPERIMENTS

**Training**     In initial pilot experiments, we calibrated the difficulty of Pathfinder and cABC challenges by identifying the smallest dataset size of the "easy" version of the challenges for which all models exceeded 90% validation accuracy. For Pathfinder this was 60K images, and for cABC this was 45K, and we used these image budgets for all difficulty levels of the respective challenges (with 10% held out for evaluation).

Our objective was to understand whether models could learn these challenges on their image budgets. We were interested in evaluating model *sample efficiency*, not absolute accuracy on a single dataset nor speed of training ("wall-time"). Thus, we adopted early stopping criteria during training: training continued until 10 straight epochs had passed without a reduction in validation loss or if 48 epochs of training had elapsed. Models were trained with batches of 32 images and the Adam optimizer. We trained each model with five random weight initializations and searched over learning rates [$1e^{-3}$, $1e^{-4}$, $1e^{-5}$, $1e^{-6}$]. We report the best performance achieved by each model (according to validation accuracy) out of the 20 training runs.

**Screening feedback computations**     Our study revealed three main findings. First, we found that human participants performed well on both Pathfinder and cABC challenges with no significant

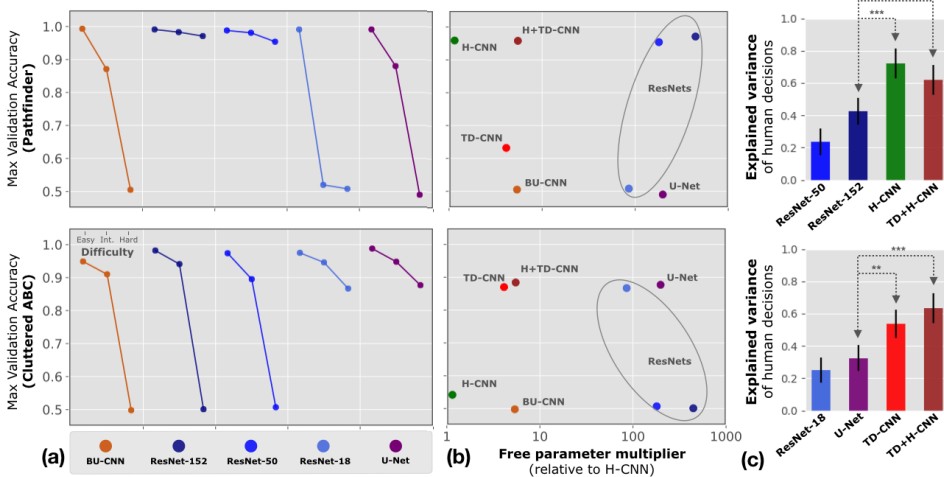

Figure 5: **Pathfinder and cABC challenges are efficiently solved by visual strategies that depend on feedback – not feedforward – processing**. **(a)** Performance of feedforward models on each difficulty of the two challenges. Deeper ResNet models solved Pathfinder, whereas the ResNet-18 and U-Net solved cABC, indicating a trade-off between capacity vs. parameter efficiency for feedforward strategies to Pathfinder vs. cABC, respectively (failures to generalize reflect overfitting). **(b)** The number of parameters of each model (plotted as a multiple of the number of parameters in the H-CNN) plotted against model performance on the hard dataset of each challenge. The H+TD-CNN solves both challenges with an order-of-magnitude fewer parameters than reference ResNet and U-Net models. **(c)** Correlations between human and leading model decisions on the hard dataset of each challenge. The TD+H-CNN and its successful lesioned variants were significantly more correlated with humans than successful feedforward reference models. Significance comparisons with the worst feedforward reference model omitted due to space constraints. P-values are derived from bootstrapped comparisons of model performance (Edgington, 1964). Error bars depict SEM; *: $p < 0.05$; **: $p < 0.01$.

drop in accuracy as difficulty increased (Fig. 4a). The fact that each challenge has been designed to feature a single cue for particular grouping (e.g., local Gestalt for Pathfinder and global semantics for cABC) suggests that human participants were able to utilize a different grouping routine for each challenge. Second, the pattern of accuracies from our recurrent architectures reveal complementary contributions of horizontal vs. top-down connections (Fig. 4b). The H-CNN solved the Pathfinder challenge with minimal straining as difficulty increased, but it struggled on the easy cABC dataset. On the other hand, the TD-CNN performed well on the entire cABC challenge but struggled on the Pathfinder challenge as difficulty increased. Together, these results suggest that top-down interactions (from higher-to-lower layers) help process object-level grouping cues, whereas horizontal feedback (between units in different feature columns/spatial locations in a layer) help process local Gestalt grouping cues. Our recurrent architecture which possessed both types of feedback, the TD+H-CNN, was able to solve both challenges at all levels of difficulties, roughly matching human performance.

All three of these recurrent models relied on recurrence to achieve their performance on these datasets. In particular, an H-CNN with one processing timestep could not solve Pathfinder, and a TD-CNN with one processing timestep could not solve cABC (Fig. S7). Separately, performance of the TD+H-CNN was relatively robust to different model configurations, and deeper versions of the model still performed well on both challenges (Fig. S8).

We also tested models on two controlled variants of the cABC challenge: one where the two letters never touch or overlap ("position control") and another where the two letters are rendered in different pixel intensities ("luminance control"). No significant straining was found on these tasks for any of our networks, confirming our initial hypothesis that local ambiguities and occlusion between the letters makes top-down feedback important for grouping on cABC images (Fig. S5).

In contrast to our recurrent architectures, its strictly-feedforward variant, the BU-CNN, struggled on both challenges, demonstrating the importance of feedback for perceptual grouping (Fig. 5a). Of the deep feedforward networks we tested, only the ResNet-50 and ResNet-152 were able to solve the entire Pathfinder challenge, but these same architectures failed to solve cABC. In contrast, the ResNet-18 and U-Net solved the cABC challenge, but strained on difficult Pathfinder datasets. Overall, no feedforward architecture was general enough to learn both types of grouping strategies efficiently, suggesting that these networks face a trade-off between Pathfinder vs. cABC performance.

We explored this tradeoff faced by ResNets in two ways. First, we tested whether the big ResNets which failed to learn the hard cABC challenge could be rescued with larger training datasets. We found that the ResNet-50 but not ResNet-152 was able to solve this task when trained on a dataset that was $4\times$ as large, demonstrating that ResNets are very sensitive to overfitting on cABC (and that the ResNet-152 needs more than $4\times$ as much data as a ResNet-18 to learn cABC). Second, we tested whether the ResNet-18, which failed to learn the hard Pathfinder challenge, could be rescued by increasing its capacity. Indeed, we found that a wide ResNet-18, with a similar number of parameters as a ResNet-50, was able to perform better (although still far worse than the ResNet-50) on the hard Pathfinder challenge (Fig. S8).

**Modeling biological feedback**  Only the H+TD-CNN and human observers solved all levels of the Pathfinder and cABC challenges. To what extent do the visual strategies learned by the H+TD-CNN and other reference models in our experiments match those of humans for solving these tasks? We investigated this with a large-scale psychophysics study, in which human participants categorized exemplars from Pathfinder or cABC that were also viewed by the models (see Appendix D for experiment details).

We recruited 648 participants on Amazon Mechanical Turk to complete a web-based psychophysics experiments. Participants were given between 800ms-1300ms to categorize each image in Pathfinder, and 800ms-1600ms to categorize each image in cABC (324 per challenge). Each participant viewed a subset of the images in a challenge, which spanned all three levels of difficulty. The experiment took approximately 2 minutes, and no participant viewed images from more than one challenge.

We gathered 20 human decisions for every image in the Pathfinder and cABC challenges, and measured human performance on each image as a logit of the average accuracy across participants (using average accuracy as a stand-in for "human confidence"). Focusing on the hard-difficulty dataset, we used a two-step procedure to measure how much human decision variance each model explained. First, we estimated a ceiling for human inter-rater reliability. We randomly (without replacement) split participants for every image into half, recorded per-image accuracy for each group, then recorded the correlation between groups. This correlation was then corrected for the split-half procedure using a spearman-brown correction (Spearman, 1910). By repeating this procedure 1,000 times we built a distribution of inter-rater correlations. We took the $95^{th}$ percentile score of this distribution to represent the ceiling for inter-rater reliability. Second, we recorded the correlation between each model's logits and human logits, and calculated explained variance of human decisions as $\frac{model}{human_{ceiling}}$.

Both the TD+H-CNN and the H-CNN explained a significantly greater fraction of human decision variance on Pathfinder images than the 50- and 152-layer ResNets, which were also able to solve this challenge (Fig. 5c; H-CNN > ResNet 152: $p < 0.001$; H-CNN > ResNet 50: $p < 0.001$; TD+H-CNN > ResNet 152: $p < 0.01$; TD+H-CNN > ResNet 50: $p < 0.001$; p-values derived from a bootstrap test, Edgington 1964). We also computed partial correlations between human and model scores on every image, which controlled for model accuracy in the measured association. These partial correlations also indicated that human decisions were significantly more correlated with the TD+H-CNN and H-CNN than they were with either ResNet model (**Explained variance/Partial correlation**; H-CNN: 0.721/0.487; TD+H-CNN: 0.618/0.425; ResNet-152: 0.426/0.290; ResNet-50: 0.234/0.159).

We repeated this analysis for the cABC challenge. Once again, the recurrent models explained a significantly greater fraction of human decisions than either the feedforward U-Net or ResNet 18 that also solved the challenge (Fig. 5c; TD-CNN > U-Net: $p < 0.01$; TD-CNN > ResNet-18: $p < 0.001$; TD+H-CNN > U-Net: $p < 0.001$; TD+H-CNN > ResNet-18: $p < 0.001$). Partial correlations also reflected a similar pattern of associations (**Explained variance/Partial correlation**; TD+H-CNN: 0.633/0.437; TD-CNN: 0.538/0.376; U-Net: 0.324/0.190; ResNet-18: 0.249/0.176).

## 6  DISCUSSION

Perceptual grouping is essential for reasoning about the visual world. Although it is known that bottom-up, horizontal and top-down connections in the visual system are important for perceptual grouping, their relative contributions are not well understood. We directly tested a long-held theory related to the role of horizontal vs. top-down connections for perceptual grouping by screening neural network architectures on controlled synthetic visual tasks. Without specifying any role for feedback connections *a priori*, we found a dissociation between horizontal vs. top-down feedback connections which emerged from training network architectures for classification. Our study provides direct computational evidence for the distinct roles played by these cortical mechanisms.

Our study also demonstrates a clear limitation of network models that rely solely on feedforward processing, including ResNets of arbitrary depths, which are strained by perceptual grouping tasks that involve cluttered visual stimuli. Deep ResNets performed better on the Pathfinder challenge, whereas the shallower ResNet-18 performed better on the cABC challenge. With more data, deep ResNets performed better on cABC; with more width, the ResNet-18 performed better on Pathfinder. We take this pattern of results as evidence that the architectures of feedforward models like ResNets must be carefully tuned on complex visual tasks. Because Pathfinder does not have semantic cues, it presents a greater computational burden for feedforward models than cABC, making it a better fit for very deep feedforward architectures; because cABC has relatively small datasets, deeper ResNets were quicker to overfit. The highly-recurrent TD+H-CNN was far more flexible and efficient than the reference feedforward models, and made decisions on Pathfinder and cABC images that were significantly more similar to those of human observers. Our study thus adds to a growing body of literature (George et al., 2017b; Nayebi et al., 2018b; Linsley et al., 2018b;a; Kar et al., 2019) which suggests that recurrent circuits are necessary to explain complex visual recognition processes.

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

## A  CLUTTERED ABC

The goal of the cluttered ABC (cABC) challenge is to test the ability of models and humans to make perceptual grouping judgements based solely on object-level semantic cues. Images in the challenge depict a pair of letters positioned sufficiently close to each other to ensure frequent overlap. To further minimize local image cues which might permit trivial solutions to the problem, we use fonts as letter images that contain uniform and regular strokes with no distinct decorations. Two publicly available fonts are chosen from the website `https://www.dafont.com`: "Futurist fixed-width" and "Instruction".

Random transformations including affine transformations, geometric distortions, and pixelation are applied before placing each letter on the image to further eliminate local category cues. Positioning and transformations of letters are applied randomly, and their variances are used to control image variability in each difficulty level of the challenge.

**Letter transformations**   Although each letter in the cABC challenge is sampled from just two different fonts, we ensure that each letter appear in sufficiently varied forms by applying random linear transformations to each letter. We use three different linear transformations in our challenge: rotation, scaling, and shearing. Each type of transformation applied to an image uses three different sets of parameters: two sets that are sampled and applied separately to each letter, and an additional one one that is sampled once commonly applied to both letters. We call the first type of transformation parameters "letter-wise" and the second type "common" transformation parameters. For example, rotations applied to the first and the second letter are each described by the sum of letter-wise and common rotational parameters, $\phi_1 + \phi_c$ and $\phi_2 + \phi_c$, respectively. Scaling is described by the product of letter-wise and common scale parameters, $S_1 S_c$ and $S_2 S_c$. Shearing is described by the sum of letter-wise and common shear parameters, $E_1 + E_c$ and $E_2 + E_c$. Each shear parameter specifies the value of the off-diagonal shear matrix applied to each letter image. Either vertical or horizontal shear is applied to both letters in each image with shear axis randomly chosen with equal probability. The random transformation procedure is visually detailed in Fig. S1.

We decompose each transformation to letter-wise and common transformation to independently increase variability at the level of an image while keeping letter-wise variability constant. This is achieved by gradually increasing variance of the letter-wise transformation parameters while decreasing variance of the common parameters. See S1 for a summary of the distributions of random transformation parameters used in the three levels of difficulty.

We apply additional, nonlinear transformations to letter images. First, geometric warping is applied independently to each affine-transformed letter image by computing a "warp template" for each image. A warp template is an image in same dimensions as each letter image, consisting of 10 randomly placed Gaussians with sigma (width) of 20 pixels. Each pixel in a letter image is translated by a displacement proportional to the gradient of the warp template in the corresponding location. This process is depicted in Fig. S2. Second, the resulting letter image is "pixelated" by first partitioning the letter image into a grid of 5×5 pixels. Each grid is filled with 255s (white) if and only if more than 30% of the pixels are white, resulting in a binary letter image in $\frac{1}{5}$ of the original resolution. The squares then undergo random translation with displacement independently sampled for each axis. We use normal distribution with standard deviation of 2 pixels truncated at two standard deviations.

**Letter positioning**   We sample the positions of letters by first defining an invisible circle of radius $r$ placed at the center of the image. Two additional random variables, $\theta$ and $\Delta\theta$, are sampled to specify two angular positions of the letters on the circle, $\theta$ and $\theta + \Delta\theta$. $r$, $\theta$ and $\Delta\theta$ are sampled from uniform distributions whose ranges increases with difficulty level (Table S1). By sampling these parameters from increasingly larger intervals, we increase relative positional variability of letters in harder datasets. In easy difficulty, we choose $r \sim U(50, 60)$. In intermediate and hard difficulties, we choose $R \sim U(40, 70)$ and $R \sim U(30, 80)$, respectively. For $\theta$ and $\Delta\theta$, we chose $\theta_1 \sim U(-30, 30)$ and $\Delta\theta \sim U(170, 190)$ for the easy difficulty, $\theta_1 \sim U(-60, 60)$ and $\Delta\theta \sim U(110, 250)$ for the intermediate difficulty, and $\theta_1 \sim U(-90, 90)$ and $\Delta\theta \sim U(70, 290)$ for the hard difficulty.

Letter images are placed on the sampled positions by aligning the positions of each letter's "center of mass", which is defined by the average (x, y) coordinate of all the foreground pixels in each letter image, with each of the sampled coordinates in an image.

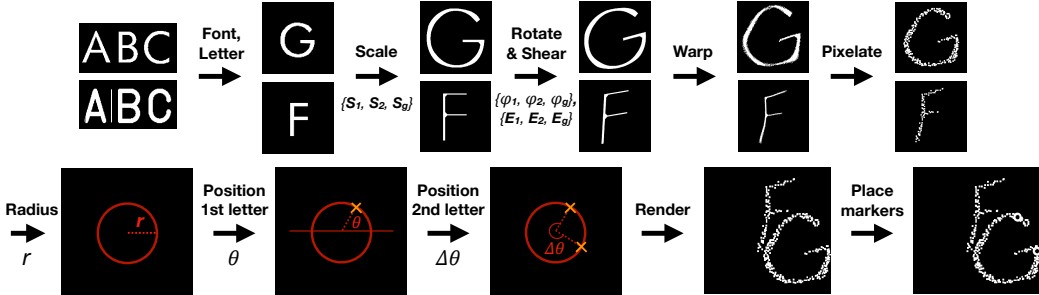

Figure S1: Visual depiction of the cABC image generation algorithm. **(Top row)** Generation procedure of individual letter images. Two independently sampled letter categories are rendered in a randomly sampled (but same) font. Each letter image undergoes three linear transformations – scale, rotation and shear – as well as two steps of nonlinear transformations – warp and pixelation. Transformation magnitudes are determined by random variables whose distributions differ between levels of difficulties. **(Bottom row)** Positional sampling procedure. Two letters are placed on two randomly chosen points on an invisible circle of radius $r$. Two angular positions on the circle are sampled by first sampling the position of the first letter, $\theta$, and then sampling the *angular displacement* of the second letter relative to the first letter, $\Delta\theta$. Each letter is positioned by aligning its "center of mass", which is the average coordinate of white pixels in a letter image, with the coordinate of the sampled position on the circle.

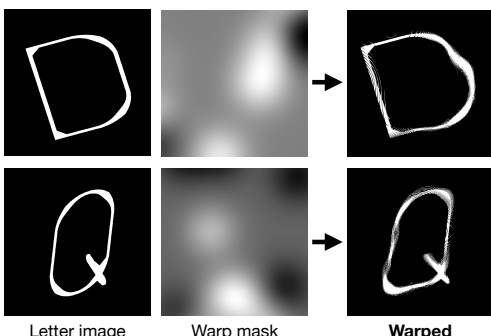

Letter image    Warp mask    **Warped**

Figure S2: Two example letter images undergoing geometric warping. A "warp template" for each image is sampled by randomly overlaying 10 Gaussian images. Each pixel in a letter image is translated by a displacement proportional to the gradient of its warp template in the corresponding location.

| Parameters | Easy | Intermediate | Hard |
|---|---|---|---|
| $\theta$ (degrees) | $U(-30, 30)$ | $U(-60, 60)$ | $U(-90, 90)$ |
| $\Delta\theta$ (degrees) | $U(170, 180)$ | $U(110, 180)$ | $U(70, 180)$ |
| $r$ (pixels) | $U(25, 30)$ | $U(20, 35)$ | $U(15, 40)$ |
| $\phi_1, \phi_2$ (degrees) | 0 | $\mathcal{N}(0, \frac{30}{\sqrt{2}})$ | $\mathcal{N}(0, 30)$ |
| $\phi_c$ (degrees) | $\mathcal{N}(0, 30)$ | $\mathcal{N}(0, \frac{30}{\sqrt{2}})$ | 0 |
| $S_1, S_2$ | 1 | $log\mathcal{N}(1.5, 0, \frac{1}{2\sqrt{2}})$ | $log\mathcal{N}(1.5, 0, \frac{1}{2})$ |
| $S_c$ | $log\mathcal{N}(1.5, 0, \frac{1}{2})$ | $log\mathcal{N}(1.5, 0, \frac{1}{2\sqrt{2}})$ | 1 |
| $E_1, E_2$ | 0 | $\mathcal{N}(0, \frac{2}{10\sqrt{2}})$ | $\mathcal{N}(0, \frac{2}{10})$ |
| $E_c$ | $\mathcal{N}(0, \frac{2}{10})$ | $\mathcal{N}(0, \frac{2}{10\sqrt{2}})$ | 0 |

Table S1: Distributions of transformation parameters used in cluttered ABC challenge. $U(a, b)$ denotes uniform distribution with range $[a, b]$; $\mathcal{N}(\mu, \sigma)$ denotes normal distribution with mean $\mu$ and standard deviation $\sigma$; $log\mathcal{N}(z, \mu, \sigma)$ denotes log-normal distribution with base $z$, mean $\mu$ and standard deviation $\sigma$.

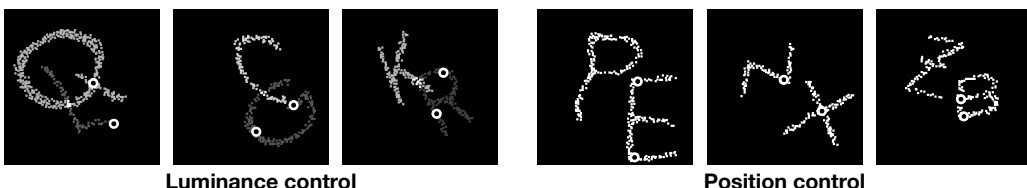

**Luminance control**  **Position control**

Figure S3: **Two control cABC challenges.** In luminance control, two letters are rendered in different, randomly sampled pixel intensity values. In positional control, two letters are always rendered without touching or overlapping each other.

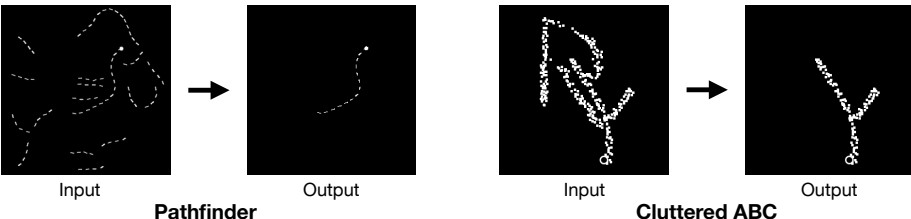

Input  Output  Input  Output
**Pathfinder**  **Cluttered ABC**

Figure S4: **Segmentation version of the two challenges.** Here, a model is tasked with producing as output an image which contains only the object which is marked in the input image.

# B  ADDITIONAL EXPERIMENTS

**Control cABC challenges**  To further validate our implementation of the cABC, we added two control cABC challenges in our study (Fig. S3). In the "luminance control" challenge, two letters are rendered with a pair of uniformly sampled random pixel intensity values that are greater than 128 but always differ by at least 40 between the two letters. In the "positional control" challenge, the letters are disallowed from touching or overlapping. Relative to the original challenge, the two controls provide additional local grouping cues with which to determine the extent of each letter. The luminance control challenge permits additional pixel-level cue with which a model can infer image category by comparing the values of letter pixels surrounding two markers. Positional control challenge provides unambiguous local Gestalt cue to solve the task. Ensuring that the two letters are spatially separated has an additional effect of minimizing the amount of clutter to interfere the feature extraction process in each letter.

In both control challenges, we found no straining effect in *any* of the models we tested. This strengthens our initial assumption behind the design of cABC that the only way to efficiently solve the default cABC challenge is to employ object-based grouping strategy. Because all networks successfully solved the control challenges, the relative difficulty suffered by the H-CNN or reference

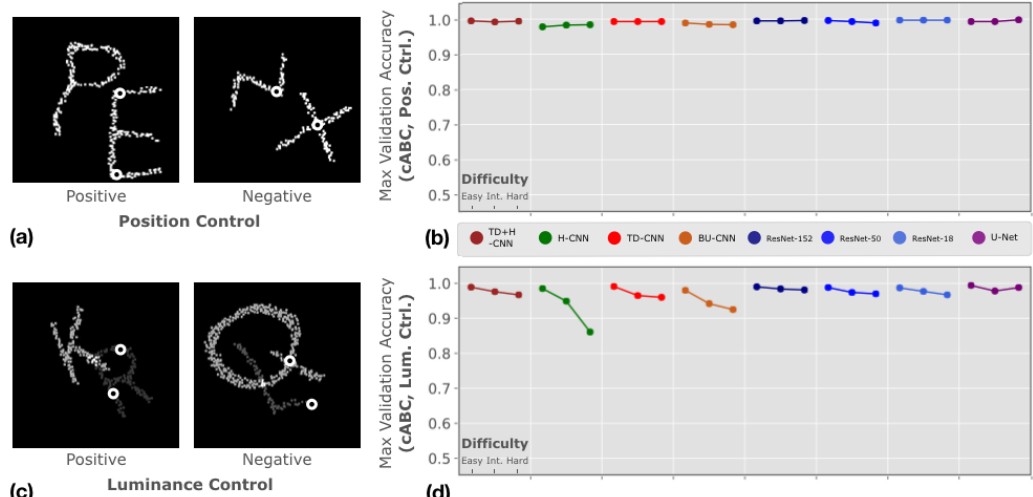

Figure S5: **Performance on two cABC-control challenges.** Top-down feedback is critical when local grouping cues are unavailable. We verify this by constructing two controlled versions of the cABC challenge, where local Gestalt grouping cues are provided in the form of spatial segregation between two letters (**a**), or in the form of luminance segregation (**b**). In both cases, the architectures that lack top-down feedback mechanism can solve the challenge at all levels of difficulties.

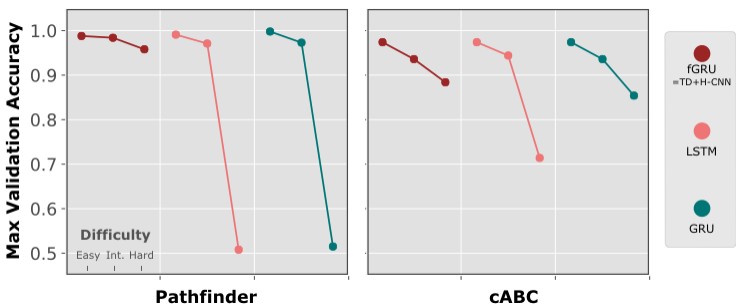

Figure S6: **fGRUs outperform GRUs and LSTMs on Pathfinder and cABC.** Our recurrent hierarchical architectures performed better on Pathfinder and cABC challenges when they used fGRUs instead of either LSTMs (Hochreiter & Schmidhuber, 1997) or GRUs (Cho et al., 2014a).

feedforward networks on the original cABC couldn't have been caused by idiosyncracies of image features we used such as the average size of letters. These networks lack the mechanisms to separate the mutually occluding letter strokes according to high-level hypothesis about the categories of letters present in each image.

**Segmentation challenges**  To further understand and visualize the nature of computations employed by our recurrent architectures when solving the Pathfinder and the cABC challenge, we construct a variant of each dataset where we pose a segmentation problem. Here, we modify each dataset by placing only one marker per image instead of two. The task is to output a per-pixel prediction of the object tagged by the marker (Figure S4). We train each model using 10 thousand images of the cABC challenge and 40 thousand images of the Pathfinder challenge and validate using 400 images in both challenges. We only use hard difficulty in each challenge. We use only the TD- and H-CNN in this experiment. We replace the readout module in each of our recurrent architecture by two $1 \times 1$ convolution layers with bias and rectification after each layer. We also include the "reference" recurrent architectures in which the fGRU modules have been replaced by either LSTM Hochreiter & Schmidhuber (1997) or GRU Cho et al. (2014a). Because the U-Net is a dense prediction architecture by design, we simply remove the readout block that we used in the main categorization challenge.

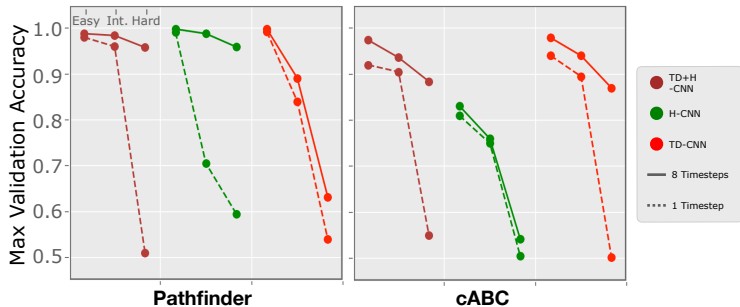

Figure S7: **Recurrence improves fGRU model performance on Pathfinder and cABC.** Each of our recurrent models performed better when given 8-timesteps of recurrent processing instead of 1 timestep.

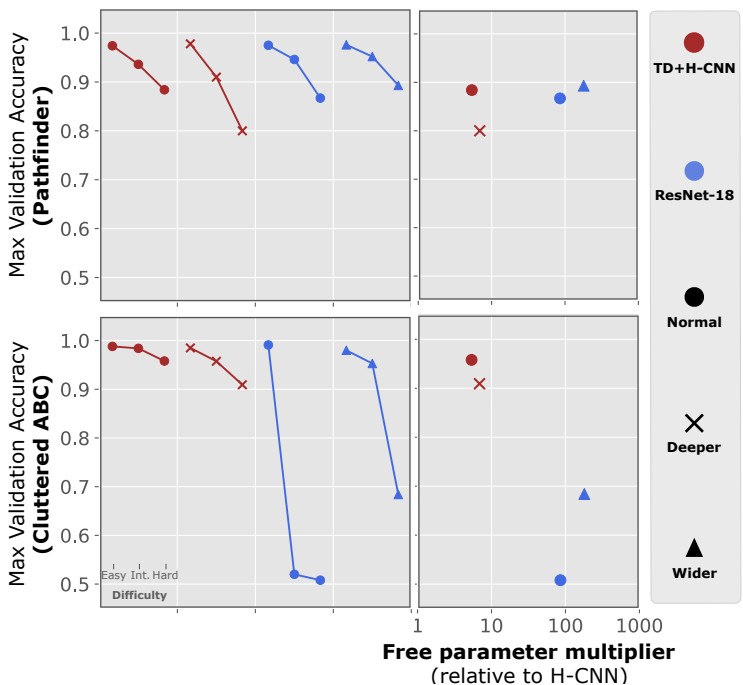

Figure S8: **A comparison of different parameterizations of H+TD-CNN and ResNet-18 models on Pathfinder and cABC. (a)** We compared the original H+TD-CNN to a deeper version (5 instead of 3 convolutional blocks between its low- and high-level fGRU modules). This deeper H+TD-CNN performed similarly to the original version. We also tested whether widening the ResNet-18 could rescue its performance on Pathfinder. Indeed, the wider ResNet-18 (matched in number of parameters to ResNet-50) performed far better on Pathfinder than the original version, but was still far worse on the task than a normal ResNet-50. **(b)** Performance of each of these models is plotted as a function of their total free parameters. Variants of the TD+H-CNN are far more parameter efficient at solving Pathfinder and cABC than variants of ResNet-18. Ellipses group variants of each model class.

We found that the target curves in the Pathfinder challenge is best segmented by H-CNN (0.79 f1 score), followed by H-CNN with GRU (0.72) and H-CNN with LSTM (0.69). No top-down-only architecture was able to successfully solve this challenge (we set a cutoff of greater than 0.6 f1). The cABC challenge is best segmented by the TD-CNN (0.83), followed by TD-CNN with LSTM (0.64) and TD-CNN with GRU (0.62). Consistent with the main challenge, no purely horizontal architecture was able to successfully solve this challenge. In short, we were able to reproduce the dissociation in performance we found between horizontal and top-down connections in other recurrent architectures (LSTM and GRU) as well, although these recurrent modules were less successful than the fGRU.

We visualize the internal update performed by our recurrent networks by plotting the difference of norm of each per-pixel feature vector at every timestep in the low-level recurrent state, $\mathbf{H}^{(1)}$ (closest to image resolution), vs. its immediately preceding timestep. Of course, this method does not reveal the full extent of the computations taking place in the fGRU, but it at least serves as a proxy for inferring the spatial layout of how activities evolve over timesteps (4b). More examples of both sucessful and mistaken segmentations and the temporal evolution of internal activities of H-CNN (on the Pathfinder challenge) TD-CNN (on the cABC challenge) are shown in Fig.S10 and Fig.S11

**Using conventional recurrent layers** We tested the performance of modifications of TD+H-CNN by replacing the fGRU modules with conventional recurrent modules – LSTMs (Hochreiter & Schmidhuber, 1997) and GRUs (Cho et al., 2014a). We found that neither of these architectures matched the performance of our fGRU-based architecture on either grouping challenges S6. The degradation of performance was most pronounced in Pathfinder. Similar to the findings in Linsley et al. (2018b), we believe that the use of both multiplicative and additive interactions as well as the separation of the suppressive and facilitatory stages in fGRU are critical for learning stable transitive grouping in Pathfinder.

## C  NETWORK ARCHITECTURES

### C.1  THE fGRU MODULE

At timestep $t$, the fGRU module takes two inputs, an instantaneous external drive $\mathbf{X} \in \mathbb{R}^{H \times W \times K}$ and a persistent recurrent state $\mathbf{H}[t-1] \in \mathbb{R}^{H \times W \times K}$ and produces as output an updated recurrent state $\mathbf{H}[t]$. Detailed description of its internal operation over a single iteration from $[t-1]$ to $[t]$ is defined by the following formulae. For clarity, tensors are bolded but kernels and parameters are not:

$$
\begin{aligned}
\mathbf{G}^I &= sigmoid(\text{BN}(U^I * \mathbf{H}[t-1])) \\
\mathbf{C}^I &= \text{BN}(W^I * (\mathbf{H}[t-1] \odot \mathbf{G}^I)) \\
\mathbf{Z} &= \left[ \mathbf{X} - \left[ (\alpha \mathbf{H}[t-1] + \mu) \mathbf{C}^I \right]_+ \right]_+ \\
\mathbf{G}^E &= sigmoid(\text{BN}(U^E * \mathbf{Z})) \\
\mathbf{C}^E &= \text{BN}(W^E * \mathbf{Z}) \\
\tilde{\mathbf{H}} &= \left[ \kappa(\mathbf{C}^E + \mathbf{Z}) + \omega(\mathbf{C}^E * \mathbf{Z}) \right]_+ \\
\mathbf{H}[t] &= (1 - \mathbf{G}^E) \odot \mathbf{H}[t-1] + \mathbf{G}^E \odot \tilde{\mathbf{H}} \\
\text{where } & \text{BN}(\mathbf{r}; \delta, \nu) = \nu + \delta \odot \frac{\mathbf{r} - \widehat{\mathbb{E}}[\mathbf{r}]}{\sqrt{\widehat{\text{Var}}[\mathbf{r}] + \eta}}.
\end{aligned}
\tag{1}
$$

The evolution of the recurrent state $\mathbf{H}$ can be broadly described by two discrete stages. During the first stage, $\mathbf{H}$ is combined with an extrinsic drive $\mathbf{X}$. Interactions between units in the hidden state are computed in $\mathbf{C}^I$ by convolving $\mathbf{H}[t-1]$ with a kernel $W^I$. These interactions are next linearly and multiplicatively combined with $\mathbf{H}[t-1]$ via the $k$-dimensional learnable coefficients $\mu$ and $\alpha$ to compute the intermediate "suppressed" activity, $\mathbf{Z}$.

The second stage involves updating $\mathbf{H}$ with a transformation of the intermediate activity $\mathbf{Z}$. Here, $\mathbf{Z}$ is convolved with the kernel $W^E$ to compute $\mathbf{C}^E$. A candidate output $\tilde{\mathbf{H}}[t]$ is calculated via the $k$-dimensional learnable coefficients $\kappa$ and $\omega$, which control the linear and multiplicative terms of self-interactions between $\mathbf{C}^E$ and $\mathbf{Z}$.

Two gates modulate the dynamics of both of these stages: the "gain" $\mathbf{G}^I[t]$, which modulates channels in $\mathbf{H}[t-1]$ during the first stage, and the "mix" $\mathbf{G}^E[t]$, which mixes a candidate $\tilde{\mathbf{H}}[t]$ with the persistent $\mathbf{H}[t-1]$ during the second stage. Both the gain and mix are transformed into the range $[0, 1]$ by a sigmoid nonlinearity. This two-stage computational structure in the fGRU captures complex nonlinear interactions between units in $\mathbf{X}$ and $\mathbf{H}$. Brackets [.] denote linear rectification. Separate

applications of batch-norm are used on every timestep, where $\mathbf{r} \in \mathbb{R}^d$ is the vector of activations that will be normalized. The parameters $\delta, \nu \in \mathbb{R}^d$ control the scale and bias of normalized activities, $\eta$ is a regularization hyperparameter, and $\odot$ is elementwise multiplication.

Learnable gates, like those in the fGRU, support RNN training. But there are multiple other heuristics that also help optimize performance. We use several heuristics to train our models, including Chronos initialization of fGRU gate biases (Tallec & Ollivier, 2018). We also initialized the learnable scale parameter $\delta$ of fGRU normalizations to 0.1, since values near 0 optimize the dynamic range of gradients passing through its sigmoidal gates (Cooijmans et al., 2017). Similarly, fGRU parameters for learning additive suppression/facilitation ($\mu, \kappa$) were initialized to 0, and parameters for learning multiplicative suppression/facilitation ($\alpha, \omega$) were initialized to 0.1. Finally, when implementing top-down connections, we incorporated an extra learnable gate, which we found improved the stability of training. Consider the horizontal activities in two layers, $\mathbf{H}^{(l)}, \mathbf{H}^{(2)}$, and the function fGRU, which implements top down connections. The introduction of this extra gate means that top-down connections are learned as: $\mathbf{H}^{(l)} = (1 - sigmoid(\beta)) \odot \text{fGRU}(\mathbf{H}^{(l)}, \mathbf{H}^{(l+1)}) + sigmoid(\beta) \odot \mathbf{H}^{(l)}$, where $\beta \in \mathbb{R}^K$ is initialized to 0 and learns how to incorporate/ignore top-down connections.

The broad structure of the fGRU is related to the Gated Recurrent Unit (GRU, (Cho et al., 2014b)). Nevertheless, one of the main aspects in which the fGRU diverges from the GRU is that its state update is carried out via two discrete steps of processing instead of one. This is inspired by a computational neuroscience model of contextual effects in visual cortex (see Mély et al. 2018 for details). Unlike that model, the fGRU allows for both additive and multiplicative combinations following each step of convolution, which potentially encourages a more diverse range of nonlinear computations to be learned (Linsley et al., 2018b;a).

## C.2 ARCHITECTURE DETAILS

Using the fGRU module, we constrcut three recurrent convolutional architectures as well as one feedforward variant. First, the H+TD-CNN (Fig. S9a) is equipped with both top-down and horizontal connections. The TD-CNN (Fig. S9b) is constructed by selectively lesioning the horizontal connections in the H+TD-CNN. The H-CNN (Fig. S9c) is constructed by selectively lesioning the top-down connections in the H+TD-CNN, essentially making recurrence only take place within the low-level fGRU. Lastly the BU-CNN (Fig. S9d) is constructed by disabling both types of recurrence, turning our recurrent archecture into a feedforward encoder-decoder network. All experiments were run with NVIDIA Titan X GPUs.

**TD+H-CNN** Our main recurrent network architecture, TD+H-CNN, is built on a convolutional and transpose-convolutional "backbone". An input image $X$ is first processed by a "Conv" block which consists of two convolutional layers of $7{\times}7$ kernels with 20 output channels and a ReLU activation applied between. The resulting feature activity is sent to the first fGRU module (fGRU$^{(1)}$), which is equipped with $15{\times}15$ kernels with 20 output channels to carry out horizontal interactions. fGRU$^{(1)}$ iteratively updates its recurrent state, $\mathbf{H}^{(1)}$. The output undergoes a batch normalization and a pooling layer with a $2{\times}2$ pool kernel with $2{\times}2$ strides which then passes through the downsampling block consisting of two convolutional "stacks" (Fig. S9a, The gray box named "DS"). Each stack consists of three convolutional layers with $3{\times}3$ kernels, each followed by a ReLU activations and a batch normalization layer. The convolutional stacks increase the number of output channels from 20 to 32 to 128, respectively, while they progressively downsample the activity via a $2{\times}2$ pool-stride layer after each stack. The resulting activity is fed to another fGRU module in the top layer, fGRU$^{(2)}$, where it is integrated with the higher-level recurrent state, $\mathbf{H}^{(2)}$. The kernels in fGRU$^{(2)}$, unlike fGRU$^{(1)}$, is strictly local with $1{\times}1$ filter size. This essentially makes fGRU$^{(2)}$ a persistent memory module without performing any spatial integration over timesteps. The output of this fGRU module then passes through the "upsampling" block which consists of two transpose convolutional stacks (Fig. S9a, The gray box named "US"),. Each stack consists of one transpose convolutional layer of $4{\times}4$ kernels. Each stack also reduces the number of output channels from 128 to 32 to 12, respectively, while it progressively upsamples the activity via a $2{\times}2$ stride. Each transpose convolutional layer is followed by a ReLU activations and a batch normalization layer. The resulting activity, which now has the same shape as the output of the initial fGRU, is sent to the third fGRU module (fGRU$^{(3)}$) where it is integrated with the recurrent state of fGRU$^{(1)}$, $\mathbf{H}^{(1)}$. This module plays the role of integrating

top-down activity (originating form "US") and bottom-up activity (originating from fGRU$^{(1)}$) via $1{\times}1$ kernels. To summarize, fGRU$^{(1)}$ in our architecture implements horizontal feedback in the first layer while the fGRU$^{(3)}$ implements top-down feedback.

The above steps repeat over 8 timesteps, and in the $8^{th}$ timestep, the final recurrent activity from fGRU$^{(1)}$, $\mathbf{H}^{(1)}$, is extracted from the network and passed through batch normalization. The resulting activity is processed by the "Readout" block which consists of two convolutional layers of $1{\times}1$ kernels with a global max-pooling layer between. Each convolutional layer has one output channel and uses ReLU activations.

**Lesioned architectures**  Our top-down-only architecture, the TD-CNN, is identical to the TD+H-CNN, but its horizontal connections have been disabled by replacing the spatial kernel of fGRU$^{(1)}$ with a $1 \times 1$ kernel, thereby preventing spatial propagation of activities within the fGRU altogether. Thus, its recurrent updates in the low-level layer are only driven by top-down feedback (in fGRU$^{(3)}$). The H-CNN is implemented by disabling fGRU$^{(3)}$. This effectively lesions both the downsampling and upsampling pathways entirely from contributing to the final output. As a result, its final category decision is determined entirely by low-level horizontal propagation of activities in the netwrok.

Lastly, we construct the purely feedforward architecture, called the BU-CNN, by lesioning both horizontal and top-down feedback. It replaces the spatial kernel in fGRU$^{(1)}$ by a $1 \times 1$ kernel and disables fGRU$^{(3)}$. By running for only one timestep, the BU-CNN is equivalent to a deep convolutional encoder-decoder network.

**Feedforward reference networks**  Because U-Nets (Ronneberger et al., 2015) are designed for dense image predictions, we attach the same readout block that we use to decode fGRU activities ($\mathbf{H}^{(1)}$) to the output layer of a U-Net, which pairs a VGG16 as its downsampling architecture, with upsampling + convolutional layers to transform its encodings into the same resolution as the input image.

# D  HUMAN EXPERIMENTS

We devised psychophysics categorization experiments to understand the visual processes underlying the ability to solve tasks that encourage perceptual grouping.

These visual categorization experiments adopted a paradigm similar to Eberhardt et al. (2016), where stimuli were flashed quickly and participants had to respond within a fixed period of time. We conducted separate experiments for the "Pathfinder" and the "cluttered ABC" challenge datasets. Participants had several possible response time (RT) widows for responding to stimuli in each dataset. For "Pathfinder", we considered $\{800, 1050, 1300\}$ ms RT windows, while for the "cluttered ABC", we consider $\{800, 1200, 1600\}$ ms. We recruited 324 participants for each experiment (648 overall) from Amazon Mechanical Turk (`www.mturk.com`).

The "Pathfinder" and the "cluttered ABC" are binary categorization challenges with "Same" and "Different" as the possible responses. For both experiments, we present 90 stimuli to each participant and assign a single RT to that participant. These 90 stimuli consist of three sets of 30, corresponding to the three level of difficulties present in the challenge datasets. We randomize the order of the three difficulties (sets of 30 stimuli) across participants as well as the assignment of keys to class "Same" and "Different". Participant accuracy is shown in Fig. 4.

**Stimulus generation**  We generated a short video for each stimuli image from the challenge dataset based on the response time allowed from the onset of stimuli. We generated three stimulus videos for each challenge image, correponding to the three different response times chosen. In each stimulus video, we included an image of a cross (black plus sign overlaid on a white background) for 1000 ms before the onset of stimulus that is shown for RT ms. The stimuli were stored as `.webm` videos and presented for 1000+RT ms during the experiments. We created a video so that the cross image and the stimulus image are seamlessly combined and shown without delays, which would otherwise be caused by loading their separate images in the browser, which introduces noise in stimulus timing.

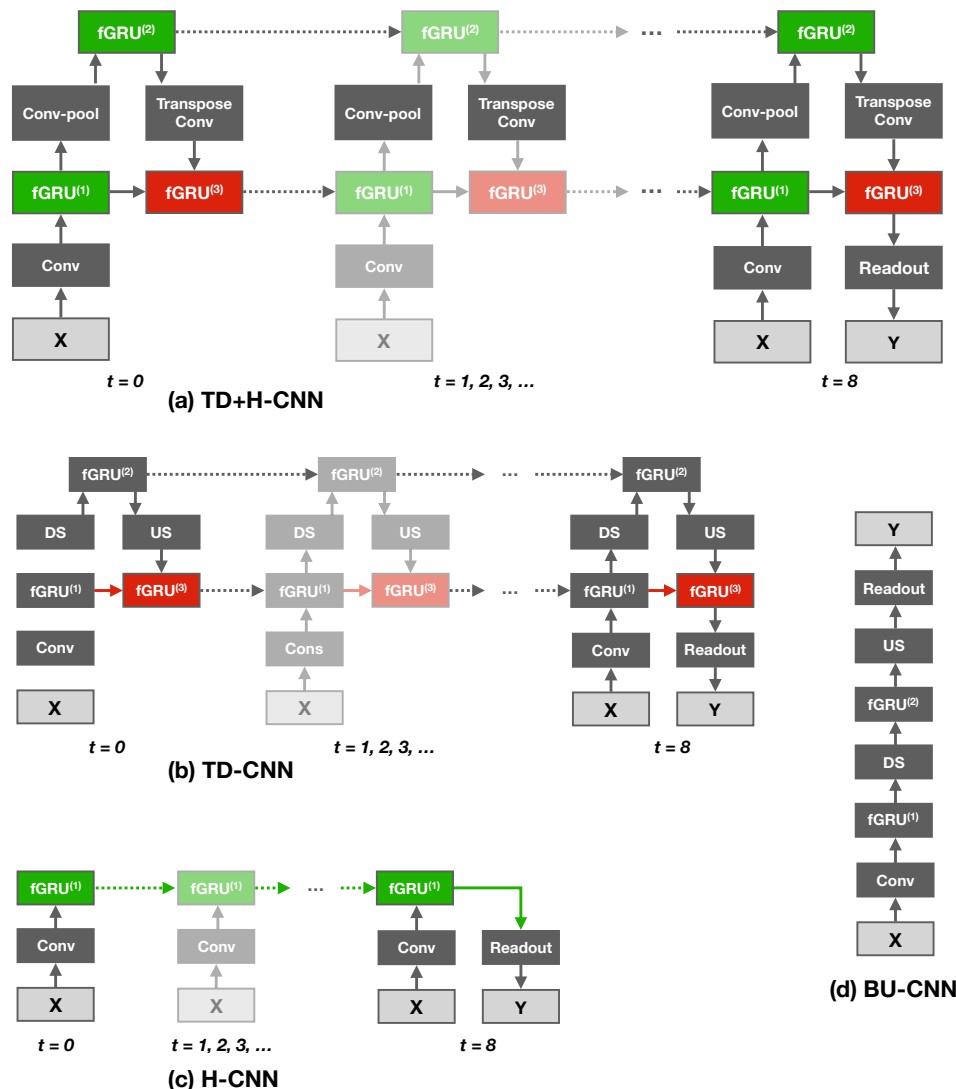

Figure S9: **Our recurrent convolutional architectures.** Feedforward or bottom-up connections (gray) pass information from lower-order to higher-order layers; horizontal connections (green) pass information between units in the same layer separated by spatial location and/or channels; top-down connections (red) pass information from higher-order to lower-order layers. We use "feedback" as an umbrella term for both top-down and horizontal connections as these can only be implemented in a model with recurrent activity that supports incremental updates to model units.

We selected a set of 200 images per difficulty in both challenges (600/challenge). Each participant responded to 30 images from each difficulty. We collected responses to the 600 images in each challenge from 18 different participants.

**Psychophysics experiment**   We conducted separate experiments for: (1) Pathfinder and (2) cABC images. The image dataset used for each experiment consisted of three difficulties - easy, intermediate and hard. Participants were assigned three blocks of trials, each having stimuli from a single difficulty level. The order of these three blocks is randomized and balanced across participants. The keys for responses to the stimuli are set to be + and −. Their assignment to "Same" and "Different" class label is randomized across participants as well.

Each experiment began with a training phase, where participants responded to 6 trials without a time limit. This phase ensured that participants understood the task and the type of image stimuli. Next,

participants went through 15 trials, 5 from each difficulty level, where they responded by pressing + or − within a limited response period. For this set of images, participants were given feedback on whether their response was correct or incorrect. After this block of "training" trials, each participant completed 90 "test" trials and responded with either "Same" or "Different" in the stipulated amount of response time, without feedback on their performance. Participants were given time to rest and the difficulty level of the viewed images changed after every 30 trials. Before starting a new block of trials, a representative image from that difficulty level was shown and instructions were given to respond as quickly as possible.

Experiments were implemented using jsPsych and custom javascript functions. We used the `.webm` format to ensure fast loading times. Videos were rendered at $256 \times 256$ pixels.

**Processing Responses**    To generate the final results from human responses on the two categorization tasks, we filtered responses rendered in less than $450$ ms, which we deem unreliable responses following the criteria of Eberhardt et al. (2016).

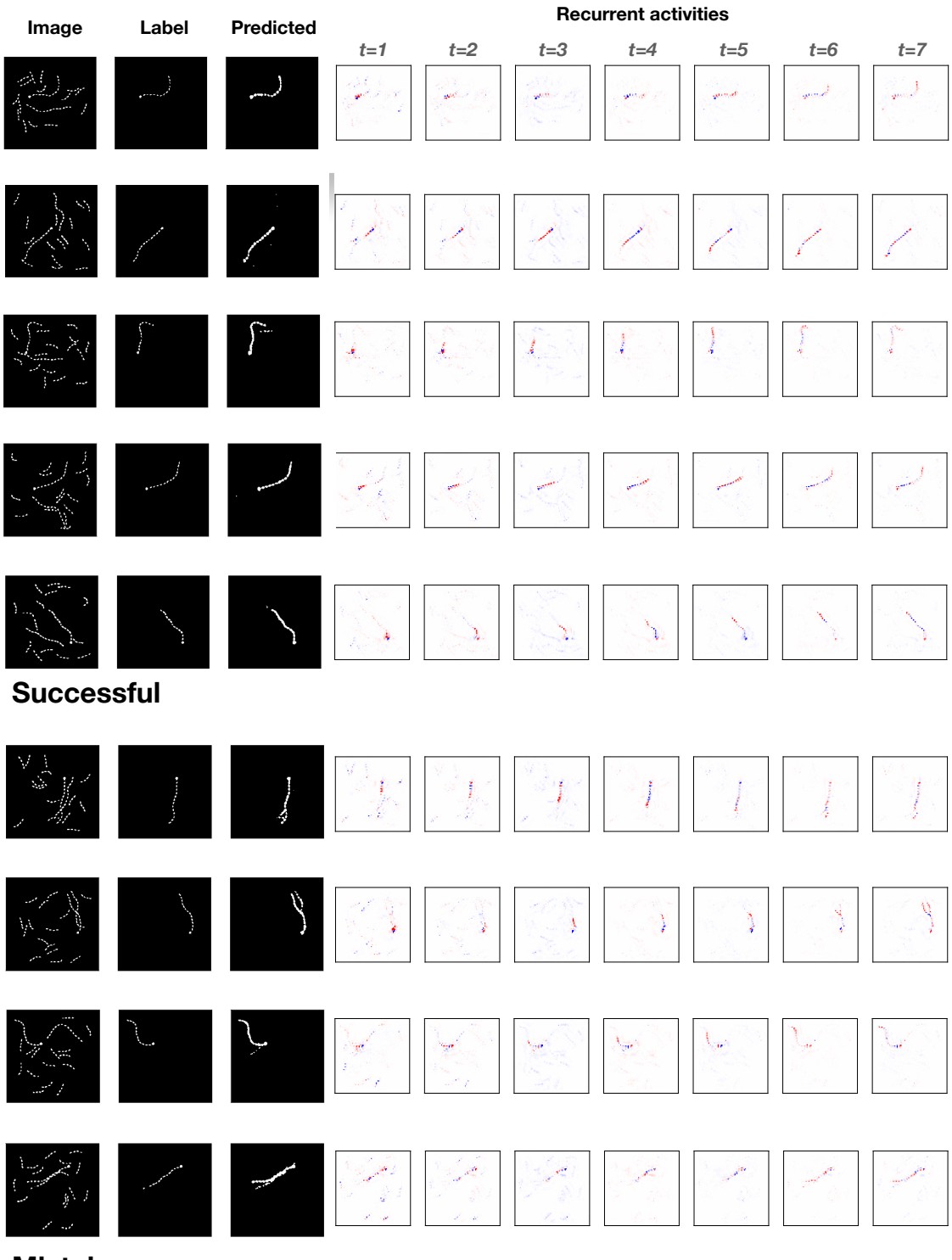

Figure S10: Activity time-courses for the Pathfinder challenge (Hard).

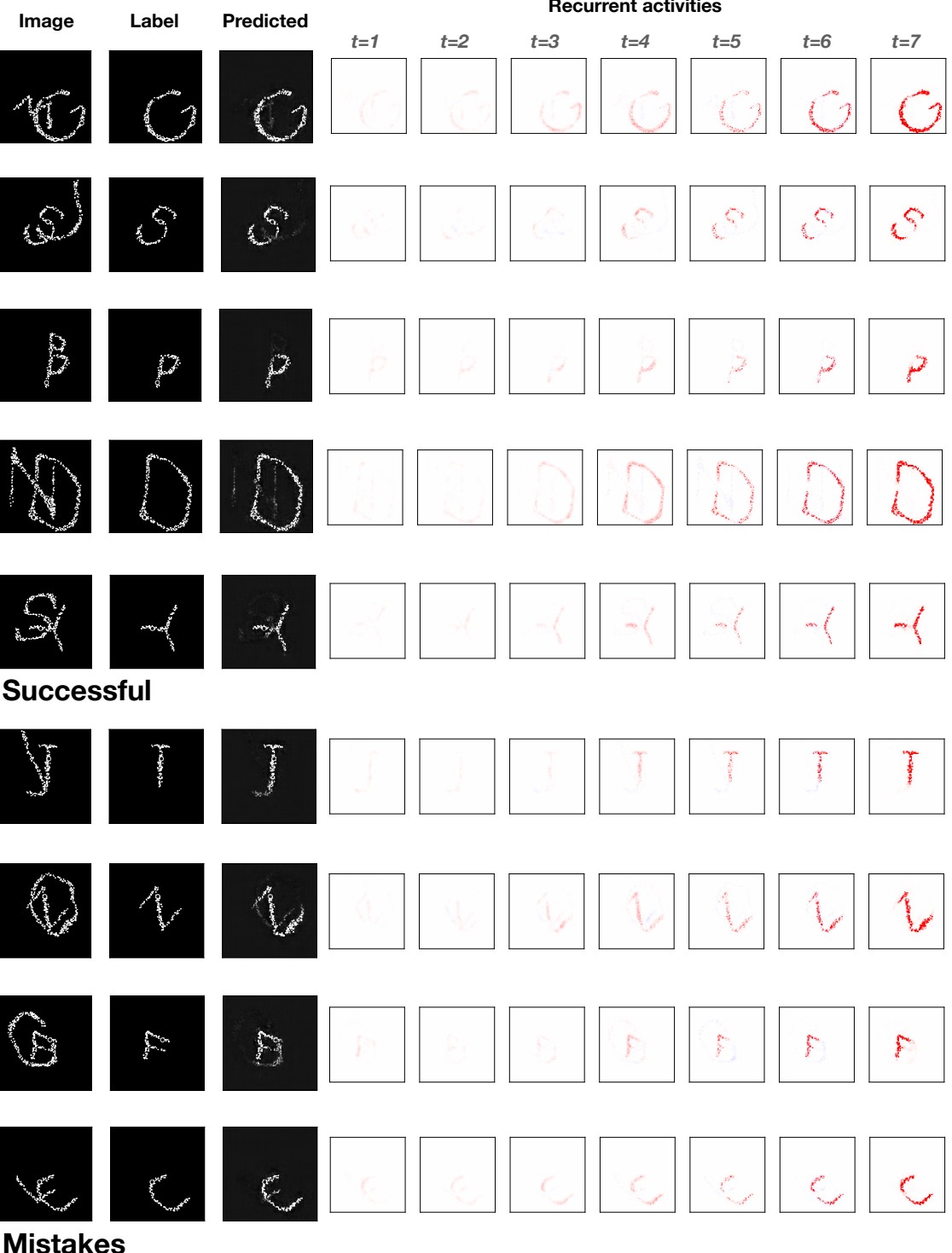

Figure S11: Activity time-courses for the cABC challenge (Hard).

