# OpenReview forum: "Disentangling neural mechanisms for perceptual grouping"
_ICLR.cc/2020/Conference — Accept (Spotlight)_

### Official Review · AnonReviewer1 · 2019-10-23
**Official Blind Review #1**

**Rating:** 8

**Review:**

The article tries to examine existing hypotheses from the neuroscience and perception literature by using neural networks as a computational model of the brain. Namely, the authors assess the efficiency of different strategies for solving two visual challenges, one of which is novel. The authors also evaluate the level of consistency between the performance of humans and different types of neural architectures.

I believe that the quality of this work is above the acceptance threshold. The results seem to clearly support the claims. The experiments are well-designed and an adequate number of baselines are provided. However, it should be mentioned that the conclusions are by no means surprising.

The following sentence should probably be fixed:
> ".. models that not learn overfit the training set."

Some questions (answering is optional):
 -  In the second paragraph of the introduction, the authors state that the two feedback mechanisms should be iterative. Can the authors provide elaborate as to why these strategies should be inherently iterative and simply applying the same model a small/finite amount of times is not enough?
 - The authors claim that the relatively low performance of ResNet-18 and U-Net on Pathfinder is due to a higher computational burden, yet the reason for the poor performance of ResNet-50 and ResNet-152  on cABC is the result of overfitting. Is there any evidence to support this distinction or are the authors simply arguing this because it is the most plausible explanation?

**Experience Assessment:**

I do not know much about this area.

**Review Assessment: Checking Correctness Of Derivations And Theory:**

N/A

**Review Assessment: Checking Correctness Of Experiments:**

I assessed the sensibility of the experiments.

**Review Assessment: Thoroughness In Paper Reading:**

I read the paper at least twice and used my best judgement in assessing the paper.

---

> ### Author Response · Authors · 2019-11-14
> **Response**
>
> Thank you for your review! We have tried to address your comments:
>
> <<In the second paragraph of the introduction, the authors state that the two feedback mechanisms should be iterative. Can the authors provide elaborate as to why these strategies should be inherently iterative and simply applying the same model a small/finite amount of times is not enough?>>
> Our claims of iterative processing were based on the qualitative results of Fig 4, S10, and S11. Based on this referee’s question, we have also included a new experiment to show that performance of our recurrent models depends on iterative processing through time. In Fig. S7, we record performance of TD+H-CNN, TD-CNN, and H-CNN models trained on Pathfinder and cABC challenges with 8-timesteps of recurrence vs. 1-timestep of recurrence. These models are not able to solve the “hard” versions of these tasks when they have only 1-timestep of processing.
>
> <<The authors claim that the relatively low performance of ResNet-18 and U-Net on Pathfinder is due to a higher computational burden, yet the reason for the poor performance of ResNet-50 and ResNet-152  on cABC is the result of overfitting. Is there any evidence to support this distinction or are the authors simply arguing this because it is the most plausible explanation?>>
> Thank you for this comment. We have included new experiments to test this claim of computational burden vs. overfitting for shallow vs. deep ResNets. First, we found that the ResNet-18 performs better on the Pathfinder challenge when it is widened and given approximately the same capacity as the ResNet-50 (although it still does not perform as well as ResNet-50; see Fig. S8). Second, we found that ResNet-50 performs better on cABC when it is trained with 4x as many unique samples as usual. The failures of ResNets on these challenges is somewhat of a diversion from the main thrust of our paper, which is to show when different forms of feedback are useful for segmentation. That said, it is notable that ResNet architectures are not as generally effective as our recurrent architectures in solving these challenges. These additional experiments, which you motivated, have helped bolster this finding.

---

### Official Review · AnonReviewer3 · 2019-10-23
**Official Blind Review #3**

**Rating:** 8

**Review:**

This paper proposed a dataset and designed a relevant network structure to analyze the function of horizontal and top-down connections for perceptual grouping. The used two datasets smartly isolate the requirements for exploiting Gestalt cues and object-based strategies. Appendix A detailed describes the cABC dataset, and the control experiments in Appendix B further validate the designing of the cABC. The proposed network flexibly integrates three types of connections and successfully solves both two challenges. The visualization results in Figure 4, S8 and S9 are insightful and also validate the intuitions. Overall the paper is clearly written and easy to follow.



Although the use of different types of connections in the proposed network is clear, I think the author should conduct more analysis for the standard networks and datasets.

In section 3, the author claims that the two challenges will cause a high computational burden for feedforward models like ResNets. However, in Figure 5, the results show that ResNet-152 and ResNet-50 could easily solve the Pathfinder. For cABC, ResNet-18 could solve and ResNet-50 and ResNet-152 will fail while the author illustrates this is due to ResNet-50 and ResNet-152 overfit to cABC. Those phenomena are quite misaligning with the arguments.

- Can we find a suitable ResNet structure that could solve both two challenges?
- Can we increase the depth of U-Net and make it solve both two challenges?
- Can we add more data or increase the difficulty for cABC and prevent the overfitting for ResNet-50 and ResNet-152?
- Can we keep increasing the difficulties of cABC and straining ResNet-18 and UNet? Does that difficulty level would also strain TD+H-CNN?
- We could also perform similar experiments on Pathfinder that increasing its difficulty for ResNet-50 and ResNet-152.

Also, if we train the TD+H-CNN on standard image datasets such as cifar-10 or ImageNet, what about the recurrent activities and the final acc?

=========================================================
After Rebuttal:

I thank the author for the response.

I hope the comments are useful for preparing a future version of this work when you have enough time.

**Experience Assessment:**

I do not know much about this area.

**Review Assessment: Checking Correctness Of Derivations And Theory:**

N/A

**Review Assessment: Checking Correctness Of Experiments:**

I carefully checked the experiments.

**Review Assessment: Thoroughness In Paper Reading:**

I read the paper thoroughly.

---

> ### Author Response · Authors · 2019-11-14
> **Response**
>
> Thank you for the comments!
>
> <<In section 3, the author claims that the two challenges will cause a high computational burden for feedforward models like ResNets. However, in Figure 5, the results show that ResNet-152 and ResNet-50 could easily solve the Pathfinder. For cABC, ResNet-18 could solve and ResNet-50 and ResNet-152 will fail while the author illustrates this is due to ResNet-50 and ResNet-152 overfit to cABC. Those phenomena are quite misaligning with the arguments. >>
> We have included new experiments that help clarify the failures of ResNets on Pathfinder and cABC tasks. First, we tested if the failure of ResNet-18 on Pathfinder was due to limitations in its capacity. Indeed, we found that a “wide” ResNet-18, with approximately the same number of weights as a ResNet-50 performed far better than the original ResNet-18 on pathfinder (not that it still does not totally solve the hard Pathfinder; Fig. S8). Second, we tested if the failure of deeper ResNets on cABC was due to overfitting. Verifying this hypothesis, we found that a ResNet-50 trained on 4x the amount of training data was able to solve the cABC challenge.
>
> <<Can we find a suitable ResNet structure that could solve both two challenges?>>
> Our new experiments make it clear that it is extremely difficult if not impossible to find a single ResNet architecture that solves both Pathfinder and cABC challenges efficiently (i.e., with limited sample training sets). While the main purpose of our work is to examine the computational role of different forms of feedback for visual segmentation, we believe we also demonstrate that purely feedforward models like ResNets are not as robust as recurrent models with horizontal and top-down feedback in solving Pathfinder or cABC challenges.
>
> <<Can we increase the depth of U-Net and make it solve both two challenges?>>
> We believe that U-Nets will experience a similar tradeoff as ResNets between performance on Pathfinder vs. cABC as the architecture is made deeper/shallower. We agree that this is an important question and could be a promising future research direction, but we did not have time to run these experiments during this rebuttal period.
>
> <<Can we add more data or increase the difficulty for cABC and prevent the overfitting for ResNet-50 and ResNet-152?>>
> This comment is addressed by our new experiments where we find ResNet-50 can learn cABC when trained on 4x as much data.
>
> <<Can we keep increasing the difficulties of cABC and straining ResNet-18 and UNet? Does that difficulty level would also strain TD+H-CNN? [Same question for Pathfinder and deep ResNets.]>>
> We believe that increasing the difficulty of either challenge will eventually make networks that rely on some type of template matching fail. That said, we also tried to constrain the parameters of our datasets so that the tasks are still trivial for humans (as shown in our psychophysical evaluation).
>
> <<Also, if we train the TD+H-CNN on standard image datasets such as cifar-10 or ImageNet, what about the recurrent activities and the final acc?>>
> The TD+H-CNN uses fGRU models, which were first introduced by [1]. They build deep hierarchical recurrent networks with fGRUs, which were trained to solve contour detection tasks in natural images and electron microscopy imaging. They found that these models are more sample efficient than state-of-the-art feedforward solutions. They also visualize the recurrent activities and find qualitatively interesting changes in processing over the model’s timecourse.
> [1] Linsley D, Kim J, & Serre T. 2019. Sample-efficient image segmentation through recurrence. ArXiv.

---

### Official Review · AnonReviewer2 · 2019-11-02
**Official Blind Review #2**

**Rating:** 6

**Review:**

This is an interesting paper. It seeks to disentangle the need for top-down and horizontal connections for grouping tasks using (a) a new synthetic dataset that seeks to evaluate one over the other, and (b) a new recurrent neural network model.


I think this is an interesting scientific question that is worth answering. It is especially useful given the context of what we know about the visual cortex in the brain, which is the presence of a large number of horizontal and top-down connections.
I think this paper takes a good first step in understanding this. I liked the problem setup and the approach of looking at accuracy at a given sample complexity rather than accuracy alone. I also liked the fact that the authors used both deeper networks and state-of-the-art baselines and corrected for the parameter count. In these respects the paper is novel and thought-provoking.

I would like the evaluation to be stronger, however. I would like to see the following experiments:
(1) How do variants of the architecture (deeper/shallower) perform under the same settings? Do the conclusions change with network depth?
(2) How do these results generalize to real-world segmentation datasets? Are both top-down and horizontal connections needed for e.g., PASCAL/COCO?

Since this is not a vision conference, I am giving this a weak accept, but I would really like to see at least an evaluation of (2).

**Experience Assessment:**

I have published in this field for several years.

**Review Assessment: Checking Correctness Of Derivations And Theory:**

N/A

**Review Assessment: Checking Correctness Of Experiments:**

I assessed the sensibility of the experiments.

**Review Assessment: Thoroughness In Paper Reading:**

I read the paper at least twice and used my best judgement in assessing the paper.

---

> ### Author Response · Authors · 2019-11-14
> **Response**
>
> Thank you for your time and feedback! To address your questions:
>
> <<How do variants of the architecture (deeper/shallower) perform under the same settings? Do the conclusions change with network depth>>
> The TD+H-CNN network is relatively shallow (see Fig. 3b), so we developed a deeper version with more convolutional layers between it’s low- and high-level fGRU modules. As shown in Fig. S8, this model performed similarly to the original TD+H-CNN on the Pathfinder and cABC challenges.
>
> <<How do these results generalize to real-world segmentation datasets? Are both top-down and horizontal connections needed for e.g., PASCAL/COCO?>>
> The fGRU modules that we use in our models were introduced in [1]. In that paper it was shown that hierarchical models with fGRUs are far more sample efficient than state-of-the-art models for object contour detection in the BSDS500 dataset and cell segmentation in electron microscopy imaging, and that the combination of top-down and horizontal connections improves performance over models with only one or the other.
>
> [1] Linsley D, Kim J, & Serre T. 2019. Sample-efficient image segmentation through recurrence. ArXiv.

---

### Author Response · Authors · 2019-11-14
**Response to reviewers**

We thank the reviewers for their thoughtful and constructive reviews. We have revised our manuscript to address the critiques that were raised. We have uploaded two versions of the new manuscript: (1) our revision, and (2) a diff between the revision and original submission. Our revision includes the following changes:

- (R1/R2/R3) Figure S8, which shows performance of a deeper version of the TD+H-CNN on cABC and Pathfinder challenges.

- (R1/R2/R3) Our new Figure S8 also demonstrates that our fGRU models are more robustly able to solve the cABC and Pathfinder challenges than ResNet models. We originally found that ResNet-50/152 solved Pathfinder but not cABC, and ResNet-18 solved cABC but not Pathfinder. We found that we can rescue ResNet-50 on cABC by training it on 4x as much data. We also found that we can improve ResNet-18 performance on Pathfinder by widening it to have approximately the same number of parameters as a ResNet-50 (its performance however still lags well behind ResNet-50). In other words, these feedforward architectures are relatively brittle, w.r.t. the success of any particular configuration, in solving cABC and Pathfinder.

- (R1/R3) We carried out an experiment (described at the end of the “Screening feedback computations” portion of Section 5) to rescue very deep ResNet (50/152-layer) performance on cABC. We find that when we quadruple the training set size, ResNet-50 is able to learn the task. We take this result as evidence ResNets are very sensitive to overfitting on cABC.

- (R1) Figure S7, which compares the performance of fGRU model variants trained with 8 timesteps vs. 1 timestep of processing. We believe that this result considered alongside the qualitative evidence of fGRU model strategies for solving cABC and Pathfinder (Figs. 4, S10, and S11), stands as strong evidence for these models utilizing iterative strategies to solve the tasks.

-(R1/R2/R3) We have included other edits to improve clarity.

We will address remaining reviewer comments directly. Please let us know if you have any other questions or concerns that we can address before the response deadline.

---

> ### Author Response · Authors · 2019-11-14
> **Clarification**
>
> The two different versions of the manuscript can be found by clicking on the "Show revisions" link. The latex diff is the most recent upload, and the revised manuscript is the second most recent upload.

---

### Decision · Program_Chairs · 2019-12-19

**Decision:**

Accept (Spotlight)

**Comment:**

All the reviewers recommend acceptance. The reviews found the paper to be interesting with substantial insights.